# Analysis of the Response of *Chlamydomonas reinhardtii* to Cobalt Ions Reveals the Protective Role of Thiols, Ascorbate, and Prenyllipid Antioxidants, and the Negative Impact of Cobalt Toxicity on Photoprotective Mechanisms

**DOI:** 10.3390/plants14223496

**Published:** 2025-11-16

**Authors:** Aylin Kökten, Beatrycze Nowicka

**Affiliations:** Department of Plant Physiology and Biochemistry, Faculty of Biochemistry, Biophysics and Biotechnology, Jagiellonian University, Gronostajowa 7, 30-387 Kraków, Poland; aylin.kokten@student.uj.edu.pl

**Keywords:** antioxidant response, *Chlamydomonas reinhardtii*, cobalt, nonphotochemical quenching of chlorophyll fluorescence, photosynthetic pigments, oxidative stress markers

## Abstract

Cobalt (Co) is an essential micronutrient for many organisms, but, at higher concentrations, it becomes harmful, primarily due to competitive interactions with other metal ions. Enzyme inhibition and disruption of nutrient homeostasis may lead to oxidative stress in Co-exposed cells. Compared to other heavy metals, such as Cd, Cu, Cr, Pb, or Ni, this element has been less studied in algae with respect to its toxicity and tolerance. Taking into account Co-induced oxidative stress and antioxidant response, the studies on algae usually did not cover a wider range of antioxidants and ROS-detoxifying enzymes monitored in one model. The aim of this study was to assess the impact of CoCl_2_ on the model green microalga *Chlamydomonas reinhardtii* from a broader perspective. We monitored algal growth, photosynthetic pigment content, the maximum quantum yield of photosystem II (F_v_/F_m_), the efficiency of nonphotochemical quenching of chlorophyll fluorescence (NPQ), and oxidative stress markers (superoxide production, lipid peroxidation). The measured antioxidants included soluble thiols, ascorbate (Asc), proline (Pro), α-tocopherol (α-Toc), and plastoquinol (PQH_2_-9). The superoxide dismutase (SOD), catalase (CAT), and ascorbate peroxidase (APX) activities were also determined. Exposure to CoCl_2_ resulted in increased levels of thiols, Asc, α-Toc, PQH_2_-9, and CAT activity. At lower concentrations of CoCl_2_, no increase in oxidative stress markers was observed, suggesting efficient antioxidant protection. On the contrary, exposure to higher concentrations of CoCl_2_ caused the inhibition of growth and chlorophyll (Chl) synthesis, as well as the reduction in the Chl *a*/Chl *b* ratio, the F_v_/F_m_ parameter, the efficiency of NPQ induction, and the levels of lipophilic antioxidants, along with an increase in lipid hydroperoxides. An interesting and novel result is the inhibitory effect of Co toxicity on state transitions in exposed algae.

## 1. Introduction

In recent centuries, heavy metals (HMs) have become significant environmental pollutants, posing a threat to ecosystems and human health. Although the term ‘heavy metals’ is considered by some scientists to be imprecise and misleading [1], it is still widely used in the literature. According to a wider definition, HMs are metals and metalloids whose density in the pure elemental state exceeds 5 g/cm^3^ (some sources give 4 g/cm^3^ as a threshold) and which display toxicity to living organisms. These elements occur in rocks and may be released into the environment due to natural processes and anthropogenic activity [2]. In water and soil, HMs occur in the form of cations and oxyanions. The mobility of HM ions depends on the ion type and the chemical properties of the environment (pH, the presence of other ions, and organic compounds). High mobility results in the high bioavailability of certain HM ions. HMs include both essential micronutrients, showing toxicity at concentrations that exceed the tolerance range of a given species, as well as elements that do not play any positive role in biochemistry [2]. HMs toxicity is a complex phenomenon due to the pleiotropic effects they exert on living cells. The four main modes of HMs toxicity are as follows: (1) high affinity for crucial chemical groups, such as the thiol, histidyl, and carboxyl groups; (2) displacement of essential metal cations; (3) similarity to phosphate; (4) generation of reactive oxygen species (ROS) by autooxidation and Haber–Weiss cycling [3,4,5,6]. The most common mechanism of toxicity is due to the similarity of given HM ions with essential metal ions. This similarity allows them to replace essential metal ions in proteins and other compounds of biological importance, such as chlorophyll (Chl). In addition, competition of HM ions with essential metals for their transport systems causes a nutrient shortage and disrupts ion and water balance [2,7,8]. Highly toxic HM ions often exhibit a high affinity for thiol groups in proteins and low-molecular-weight compounds, such as glutathione (GSH) [4,5]. The ions of the so-called redox-active HMs undergo a redox cycling in cells that leads to increased ROS production [6]. However, an increase in ROS formation can also occur in response to redox-inactive HMs, as a result of general disturbance of cellular metabolism and malfunctioning of ROS-detoxifying processes [3]. The toxicity of HMs for higher plants and algae has been the subject of intensive research [2,5,9,10,11].

Among HMs, cobalt (Co) has been less examined in terms of its toxicity and tolerance in algae compared to Cd, Pb, Cu, Cr, or Ni [3]. This element naturally occurs in the Earth’s crust minerals [12]. Among the important anthropogenic sources of Co are Cu and Ni smelting and refining, alloy manufacturing, battery production, cement industry, pigments and paints, fossil fuel combustion, industrial waste, and agricultural use of phosphate fertilisers [13,14]. In many organisms, Co is a micronutrient, present in the cobalamin ring of B_12_, and in nitrile hydratase enzymes [12]. However, it also displays toxicity, which was postulated to be the result of competitive interactions with other metal ions [15]. In higher plants, excessive Co was observed to cause growth inhibition, disturbed transport of other nutrients (P, S, Cu, Zn, Mn), as well as decrease in Fe content and chlorophyll (Chl) content [2]. The application of higher Co concentrations resulted in inhibition of important enzymes, i.e., nitrate reductase, photosystem II (PSII), and phosphoenol pyruvate carboxylase. Excess Co has also been shown to inhibit RNA synthesis and disrupt mitotic spindle formation [16].

The inhibition of enzymes and disturbance of nutrient homeostasis can result in enhanced ROS formation in Co-exposed cells and the occurrence of oxidative stress. Indeed, enhanced lipid peroxidation was observed in the haptophyte *Pavlova viridis* and in the green microalgae *Scenedesmus* sp. and *Chlorella* sp. treated with excessive CoCl_2_ [17,18]. Living organisms have evolved several mechanisms of antioxidant defence [19]. Among them, the most important are low-molecular-weight antioxidants, both hydrophilic and hydrophobic, as well as ROS-detoxifying enzymes. Among hydrophilic antioxidants, very important ones are GSH and other soluble thiols, and ascorbate (Asc). Free proline (Pro) is known for its osmo-protective function, but it also displays antioxidant action [3]. On the other hand, lipophilic antioxidants, such as tocopherols (Toc), prenylquinols, and carotenoids, are crucial for the protection of membranes and storage lipids [20]. Antioxidant enzymes include superoxide dismutases (SODs), ascorbate peroxidase (APX), catalase (CAT), GSH peroxidase, and various other enzymes, including those participating in Asc and GSH recycling [3].

Organisms also have mechanisms aimed at preventing ROS formation. In photosynthetic cells under illumination, the main source of ROS is photosynthesis, due to the photooxidative action of Chl and the electron leakage from the photosynthetic electron transfer chain. Therefore, an important protective mechanism is based on the dissipation of excessive energy absorbed by the photosynthetic apparatus. It prevents prolonged excitation of Chl, decreasing the chance of unwanted energy transfer to ^3^O_2_ and formation of harmful ^1^O_2_. It also prevents overreduction in the photosynthetic electron transfer chain and the resulting electron leakage and O_2_^•−^ formation [21,22]. In *Chlamydomonas reinhardtii*, the key photoprotective mechanism of quenching of excited states of Chl depends on the light-harvesting complex stress-related (LHCSR) proteins [23].

There are no robust data in the literature on the antioxidant response of algae to Co toxicity. The activities of CAT and SOD were measured in Co-exposed *Chlorella pyrenoidosa* and *Chlorella* sp. [17,24]. Liu et al. measured the peroxidase and CAT activity in halotolerant *Dunaliella* sp. exposed to Co [25]. Finally, Li et al. evaluated the activity of SOD, CAT, GSH peroxidase, and GSH content in *P. viridis* treated with excessive Co [18]. As various antioxidants act together to prevent oxidative damage in stressed algae, simultaneous measurements of a wider range of antioxidants and ROS-detoxifying enzymes in response to Co would bring valuable information. Furthermore, the antioxidant response has not been measured in Co-exposed *Chlamydomonas reinhardtii*, which is a model species widely used in research on HMs toxicity and tolerance [3,26].

The aim of the present study was to assess the impact of CoCl_2_ on *C. reinhardtii* from a broader perspective. We monitored algal growth, photosynthetic pigment content, maximum quantum yield of photosystem II (F_v_/F_m_), the efficiency of nonphotochemical quenching of chlorophyll fluorescence (NPQ), and oxidative stress markers (O_2_^•−^ formation and the content of lipid hydroperoxides, LOOHs). The measured antioxidants included hydrophilic compounds, such as soluble thiols, ascorbate (Asc), and proline (Pro), as well as lipophilic compounds α-tocopherol (α-Toc) and plastoquinol (PQH_2_-9). The activities of SOD, CAT, and APX were also determined.

## 2. Results

As expected, excessive Co had a negative impact on algal growth (Figure 1a) and Chl *a* + *b* content (Figure 1b). Considering measurements of OD at λ = 750 nm, the results of which can be related to the number of cells in the culture, the effect of Co addition was less pronounced after 1 week of growth than after 2 weeks. In 7-day-old cultures, the statistically significant decrease in OD compared to control was observed only for the series with the highest applied CoCl_2_ concentration, that is, 125 µM; whereas, in 14-day-old cultures, it was observed in algae exposed to 80 and 125 µM CoCl_2_. The growth of the culture containing 50 µM CoCl_2_ was slightly slower compared to that of the control, but this difference was not statistically significant (Figure 1a). The inhibitory impact of CoCl_2_ on total Chl content in algal cultures was more pronounced than the impact on culture growth. After 1 week, a statistically significant decrease in the Chl *a* + *b* content was observed for the series with 50, 80, and 125 µM CoCl_2_, while after 2 weeks it was observed for the series with 40, 50, 80, and 125 µM CoCl_2_. Slight but statistically insignificant inhibition was observed in algae exposed to 25 µM CoCl_2_ (Figure 1b). The highest concentration of CoCl_2_ applied completely inhibited the growth of the culture and caused a decrease in the Chl *a* + *b* content in the exposed algae (Figure 1).

The ratio of Chl *a* to Chl *b* was decreased compared to the control in algae exposed to 40 µM CoCl_2_ both after 1 and 2 weeks of growth, and in 2-week-old cultures treated with 50, 80, and 125 µM CoCl_2_ (Figure 2a). A similar trend, that is, a decrease in the series exposed to 40, 50, 80, and 125 µM CoCl_2_, when compared to the control, was observed for the maximum quantum yield of PS II. This time, the differences between all the above-mentioned series and the control were statistically significant after 1 and 2 weeks of growth (Figure 2c). Exposure to CoCl_2_ did not cause changes in the total carotenoids to total Chl ratio, except for the series with the highest concentration of CoCl_2_ applied measured after 2 weeks of exposure, where the decrease could be seen, compared to the control (Figure 2b).

Growth in the presence of CoCl_2_ had an impact on the induction of NPQ (Figure 3). The NPQ parameter is calculated as (F_m_ − F_m’_)/F_m’_, where F_m_ is the maximum fluorescence in the dark-adapted state, measured at the beginning of the protocol, while F_m’_ is the maximum fluorescence in algae exposed to actinic light and during the subsequent dark relaxation period. The decrease in the F_m’_ parameter compared to F_m_ results in an increase in the value of the NPQ parameter. This decrease may be due to various processes occurring in the photosynthetic apparatus.

The increase in NPQ parameter measured during actinic light exposure reflects the induction of photoprotective mechanisms of non-photochemical quenching of Chl fluorescence, mainly the qE component, induced by ΔpH gradient across thylakoid membranes [27]. When the actinic light is switched off, the ΔpH gradient rapidly dissipates, qE relaxes, and the NPQ parameter decreases. However, the transition of algae from light to darkness causes the induction of chlororespiration. In the situation of limited O_2_ supply, chlororespiration causes the reduction in PQ pool in thylakoid membranes. This is a signal for the activation of the STT7 kinase responsible for the transition from state 1 to state 2. Transition to state 2 causes a decrease in the efficiency of energy transfer from LHCII antennae to PSII and, as a result, a decrease in F_m’_ [27,28]. Therefore, the increase in the NPQ parameter in darkness reflects state transitions.

In the 1-week-old culture, the series exposed to 5 µM CoCl_2_ showed more efficient induction of NPQ during actinic light exposure compared to the control, but this effect was not observed in the 2-week-old culture (Figure 3). Exposure to higher concentrations of CoCl_2_ resulted in a decreased efficiency of NPQ induction. A statistically significant decrease in NPQ was observed compared to the control for CoCl_2_ concentrations of 40 µM and higher in 1-week-old cultures and for concentrations of 50 µM and higher in 2-week-old cultures (Figure 3). The decreased NPQ in darkness, compared to the control, was observed for CoCl_2_ concentrations of 50 µM and higher in 1-week-old cultures and for concentrations of 25 µM and higher in 2-week-old cultures (Figure 3).

Oxidative stress markers were measured for the series selected among the ones mentioned above, as indicated in Figure 4. In some series exposed to excessive CoCl_2_, the formation of O_2_^−•^ was slightly increased or decreased compared to the control, but no clear trend could be observed, suggesting that there was no dose-dependent increase in O_2_^−•^ formation in Co-exposed algae (Figure 4a). The content of LOOHs increased compared to the control in *C. reinhardtii* exposed to 80 µM CoCl_2_. For each series, there was a statistically significant increase in the content of LOOHs during culture growth (Figure 4b, significance of differences between weeks 1 and 2 not marked in the plots).

The content of the chosen hydrophilic low-molecular-weight antioxidants and the activity of the major ROS-detoxifying enzymes were measured in cultures of *C. reinhardtii* exposed to CoCl_2_ after 2 weeks (Figure 5). Exposure to excessive Co resulted in an increase in low-molecular thiols and Asc, and this increase was clearly dose-dependent (Figure 5a,b). In the case of soluble thiols, the increase was approximately 50% compared to the control; meanwhile, in the case of Asc, it was much higher (up to 4.5 times more Asc compared to the control), and a pronounced increase (about 3-fold) was observed for the lowest CoCl_2_ concentration tested in the experiment, that is, 15 µM (Figure 5a,b). The protocol used enabled the evaluation of dehydroascorbate (DHA), which is a stable product of Asc oxidation, but in all the series there were only trace amounts of DHA. No increase in Pro content was observed in Co-exposed *C. reinhardtii* (Figure 5c).

Taking into account the response to Co in terms of antioxidant enzyme activity, a dose-dependent decrease was observed for SOD activity for the entire range of CoCl_2_ concentrations tested (Figure 5d), while for APX, a dose-dependent decrease was observed for CoCl_2_ concentrations ranging from 15 to 80 µM. In algae exposed to 120 µM CoCl_2_, the activity of APX increased by approximately 50% compared to the control (Figure 5e). On the other hand, CAT activity increased by approximately 30% in cultures exposed to lower CoCl_2_ concentrations (15, 30, 50 µM) and decreased by approximately 55% in algae exposed to 120 µM CoCl_2_ compared to the control (Figure 5f).

In addition to hydrophilic antioxidants, the content of the major lipophilic antioxidants, α-tocopherol (α-Toc) and plastoquinol-9 (PQH_2_-9), was also measured in Co-exposed *C. reinhardtii* (Figure 6). As these compounds are localised in plastids, where they play a crucial role in the protection of the thylakoid membrane against lipid peroxidation, their content was normalised to both total soluble protein content and total Chl content. A clear dose-dependent pattern was observed for both compounds, regardless of the normalisation method, that is, an increase for the lowest CoCl_2_ concentrations applied and then a decrease for the higher ones (Figure 6). The increase in content was more pronounced for PQH_2_-9 compared to α-Toc, but apart from that, both compounds showed similar patterns of changes. When the prenyllipid content was normalised to total soluble proteins, a progressive increase with increasing CoCl_2_ concentration in the medium was observed in the range of 15–50 µM. Higher CoCl_2_ concentrations resulted in a progressive decrease in prenyllipid content (Figure 6a,c). When the prenyllipid content was normalised to the Chl content, an increase was observed in the range of 15–80 µM CoCl_2_ (Figure 6b,d). The applied HPLC method allowed us to measure the other isoprenoid chromanols occurring in plastids, i.e., γ-tocopherol (γ-Toc) and plastochromanol-8 (PC-8), as well as the oxidised form of PQH_2_-9, plastoquinone-9 (PQ-9). The amounts of γ-Toc and PC-8 were several times lower than the amount of α-Toc, and the observed trends were similar to those observed for α-Toc. The PQ-9 content constituted only a few percent of the total pool of PQ (PQ + PQH_2_) and did not change much in the Co-exposed algae compared to the control.

## 3. Discussion

The inhibitory effect of HM ions on growth and Chl accumulation in photosynthetic organisms is a well-known phenomenon [29,30,31,32,33,34]. This effect was observed in *C. reinhardtii* exposed to HMs, such as Cd, Cr, Cu, Hg, Ni, or Pb [35,36,37,38,39,40,41,42,43,44,45,46,47]. Co ions were shown to inhibit the growth of green microalgae *Chlorella vulgaris*, *Scenedesmus obliquus*, *C. reinhardtii*, diatoms *Phaeodactylum tricornutum*, *Nitzschia perminuta, Selenastrum capricornutum* (*Raphidocelis subcapitata*), and haptophyte *P. viridis* [18,48,49,50,51,52,53]. Chl content was decreased in Co-exposed *Chlorella* and *Scenedesmus* sp. compared to the control [17]. Experiments in which both algal growth and Chl content were monitored confirmed the inhibitory effect of Co ions on *C. pyrenoidosa, Monoraphidium minutum, S. capricornutum,* and *N. perminuta* [24,54,55]. Our previous experiments showed that the negative impact of tested HMs on Chl content in *C. reinhardtii* cultures was usually more pronounced than the impact on their OD [42,43,44]. This effect was also observed in the case of Co exposure (Figure 1). In contrast, in the haptophyte *P. viridis* exposed to 10, 20, and 50 μM CoCl_2_, the growth was inhibited, but the Chl *a* content normalised to cell number increased compared to the control [18]. In our experiment, the differences in cell numbers and Chl content between the control and Co-exposed series increased during culture growth (Figure 1). Such an effect was also observed in Co-exposed *C. pyrenoidosa*, *C. reinhardtii*, *R. subcapitata, Chlorella* sp., and *Scenedesmus* sp. [17,24,50,53]. This is similar to the response of *C. reinhardtii* to Cd and Cr observed in our previous experiments [42]. On the other hand, *C. reinhardtii* treated with Cu ions seemed to acclimate to this HM during culture growth, as the negative impact of Cu ions on growth decreased over time [42,44].

Considering the effect of HM ions on total Chl level and the ratio of Chl *a* to Chl *b*, it is known that HMs are capable of inhibiting the biosynthesis of these pigments [56,57], replacing Mg^2+^ in their molecules [58], and causing Chl degradation through oxidation [29], pheophytinization [46,59], and the induction of chlorophyllase activity [60]. The inhibitory action of Co on Chl biosynthesis has been described in the literature. Csatorday et al. postulated that this effect occurred at the steps preceding the insertion of Mg in the porphyrin ring, most likely after the formation of protoporphyrin [61]. Co was also found to inhibit 5-aminolevulenic acid (ALA) synthase, ALA dehydratase, and porphobilinogenase, similar to the effect of Ni [57]. Taking into account the substitution of Mg^2+^ by HMs, Chl *a* turned out to be more susceptible to this process than Chl *b* [59]. It was also hypothesised that, during the stress evoked by HMs, Chl *a* could be nonenzymatically oxidised to Chl *b* [62]. Both of the above-mentioned effects would result in a decrease in the Chl *a*/Chl *b* ratio, often observed in photosynthetic organisms treated with HMs [29,41,62]. In the present experiment, the decrease in the Chl *a*/Chl *b* ratio was correlated with a general decrease in total Chl content compared to the control (Figure 1 and Figure 2a). It is probable that exposure of *C. reinhardtii* to Co led to the inhibition of Chl biosynthesis, as well as to increased Chl *a* degradation.

An increase in carotenoid content was observed in some experiments with algae treated with HMs [3]. For example, an increase in the carotenoids to Chl ratio occurred in Cd- and Cr-exposed *C. reinhardtii* [43], Cu- and Cr-exposed *C. vulgaris* [63,64], Cu-exposed *Dunalliella salina* and *Dunaliella tertiolecta* [65], and Ni-exposed *Dunaliella* sp. [66]. An increase in the carotenoids to Chl *a* ratio was observed in the Co-exposed diatom *N. perminuta* [54]. However, in our experiment no such effect was observed, i.e., the ratio of carotenoids to Chl remained rather stable, except for the 2-week-old series with the highest CoCl_2_ concentration applied, where the decrease in this ratio suggested an enhanced degradation of carotenoid pigments (Figure 2b). It could be hypothesised that, in our model, the accumulation of carotenoids did not play a key role during the response to Co exposure.

The decrease in the maximum quantum yield of PSII in response to HMs was observed in some experiments, for example, in spinach-derived thylakoids exposed to Cu, Hg, and Pb ions [67], as well as in Pb-exposed *Nitzschia closterium* and *C. reinhardtii* [39,68]. However, in other models, no decrease in this parameter was observed in response to HMs. No differences were observed between the control series and *C. reinhardtii* grown in the presence of Cu, Cd, Cr, Hg, and Ag ions [42]. In the case of exposure to Co, a dose-dependent decrease in the F_v_/F_m_ parameter was demonstrated for *S. capricornutum* [53,55]. This effect was also observed in our experiment, where a decrease in F_v_/F_m_ was well correlated with the decrease in total Chl content and the Chl *a*/Chl *b* ratio in Co-exposed algae, suggesting damage to the photosynthetic apparatus (Figure 1 and Figure 2a,c).

The improvement in the efficiency of nonphotochemical quenching of Chl fluorescence was observed in *C. reinhardtii* exposed to Cu, Cd, and Cr ions for 2 weeks compared to the control [42], and in the Cu-tolerant strain of *C. reinhardtii* compared to the non-tolerant strain [69]. We wanted to determine whether such an effect occurs in *C. reinhardtii* grown in the presence of increased concentrations of CoCl_2_, but enhanced induction of NPQ compared to the control was observed only for a 1-week-old culture exposed to 5 μM CoCl_2_ (Figure 3). In cultures grown in the presence of CoCl_2_ concentrations that inhibit Chl accumulation in cultures, and cause the decrease in both the Chl *a*/Chl *b* ratio and the maximum quantum yield of PSII, the decrease in the efficiency of NPQ could be seen, suggesting a general malfunction of the photosynthetic apparatus and its regulatory mechanisms. It may be speculated that the decrease in the photoprotective mechanisms would result in PSII damage and the decrease in the Chl *a* content. A decrease in NPQ efficiency was also observed in Co-exposed *S. capricornutum* [55]. An interesting and new observation is the change in the response of the NPQ parameter in darkness in cultures exposed to high CoCl_2_ concentrations compared to the control, suggesting the inhibition of dark-induced state transitions (Figure 3). More detailed studies are now required to determine the exact biochemical mechanism of this inhibition. There are some possible explanations. First, it can be hypothesised that Co directly inhibits the *stt7* kinase responsible for the phosphorylation of LHCII antennae. The second hypothesis to be tested is whether the inhibition by Co is indirect. This could occur by inhibition of the chlororespiratory electron transport chain or by significant disturbance of photosynthesis, so that light exposure would not allow cell to produce enough NADPH to fuel chlororespiration after transition to darkness.

The enhancement of ROS production is often observed in photosynthetic organisms that suffer from HM-induced stress. There are robust data in the literature that confirm an increase in various markers of oxidative stress in algae exposed to HM ions [3]. Taking into account *C. reinhardtii*, an increase in thiobarbituric acid reactive substances (TBARSs), which are markers of lipid peroxidation, was observed in response to Hg, Cu, and Ni [35,40,41,44,47,70,71]. An increase in O_2_^−•^ formation occurred in *C. reinhardtii* growing in the presence of Cd and Cr ions [43]. Exposure to Co led to an increase in TBARS in *P. viridis*, *Scenedesmus* sp., and *Chlorella* sp. [17,18]. On the other hand, in *S. capricornutum* ROS production, measured fluorometrically, increased only in the series exposed to 0.5 mg of Co (applied as CoCl_2_) dm^−3^, while there were no statistically significant differences between the control and the series containing 0.1, 0.25, and 0.75 mg of Co dm^−3^ [53]. In the present experiment, the formation of O_2_^−•^ in Co-exposed *C. reinhardtii* was similar to that of the control (Figure 4a), suggesting efficient protection against this type of ROS for a tested range of CoCl_2_ concentrations. Taking into account lipid peroxidation, an increase in the amount of LOOHs was observed compared to the control in the series with 80 μM CoCl_2_, the concentration for which significant growth inhibition occurred (Figure 1a and Figure 4b). It suggests that severe poisoning with Co ions led to the exhaustion of antioxidant protection, resulting in the progression of lipid peroxidation.

The antioxidant response to HM-induced stress has been the subject of intensive research. An enhancement of antioxidant protection is often observed in organisms exposed to HMs, although there are no universal trends. The antioxidant response depends on many factors, i.e., HM type and concentration applied, the timing of dosage, culture conditions, species examined, internal concentration of HM ions, sometimes even the type of salt used in the experiment (for example sulphate vs. chloride). This variety of responses makes it difficult to formulate general trends [3]. For example, *Scenedesmus vacuolatus* accumulated more Cu and displayed a more pronounced antioxidant response compared to *Chlorella kessleri*, which was less tolerant to Cu. Therefore, it was speculated that better protection from ROS was the reason for increased tolerance [72]. On the other hand, when tolerance, metal accumulation, and antioxidant response were assessed in Cu-exposed *Scenedesmus acuminatus* and *Chlorella sorokiniana*, the *Scenedesmus* species was found to be more tolerant than the *Chlorella* species; however, it accumulated less Cu and its antioxidant response was either similar to or much lower than that of less tolerant species. This time, the limited entry of Cu into cells seemed to be the reason for the increased tolerance of *S. acuminatus* and there was no need to up-regulate antioxidant protection [73]. Sometimes, big differences in HM tolerance are observed between closely related strains and species [74].

However, some common trends were observed. The relatively common type of response is an increase in antioxidant examined for lower doses of HM salt and a decrease for higher concentrations applied [3]. This pattern was observed for APX, SOD, and CAT activity in Hg-exposed *C. reinhardtii* [35]. The activity of the enzymes mentioned above was also increased in 1-week-old *C. reinhardtii* exposed to 20 μM CuSO_4_, which was correlated with the appearance of symptoms of oxidative stress in these algae [44]. An increase in APX activity was observed in Cr- and Cd-treated *C. reinhardtii*, while CAT activity increased only in algae exposed to Cd, and SOD activity increased in series with Cr ions and decreased in series with Cd ions [43]. A dose-dependent increase in SOD and CAT activity was observed in Co-exposed *Chlorella* and *Scenedesmus* sp. [17]. A dose-dependent increase in CAT, but not in SOD activity, was observed in *P. viridis*. In the study on *P. viridis*, the activity of GSH peroxidase was also shown to increase in Co-exposed cells [18]. On the other hand, a dose-dependent decrease in SOD and CAT activity has been reported in Co-exposed *C. pyrenoidosa* [24]. In the present experiment, exposure to Co led to a dose-dependent decrease in SOD activity, suggesting the inhibitory action of Co^2+^ on SOD activity in our model (Figure 5d). *C. reinhardtii* contains two types of SOD, MnSOD and FeSOD [3]. Co was shown to be capable of replacing Mn in MnSOD of *E. coli* and the substituted enzyme was not active [75]. This metal may also replace the metal ion at the active site of the cambialistic Fe/Mn SOD, related both to MnSOD and FeSOD [76]. Therefore, it is plausible that the decrease in SOD activity is caused by an inhibition of an enzyme. This topic requires a study focused on the impact of Co on the activity of MnSOD and FeSOD and the expression of their genes in *C. reinhardtii*, to be carried out in the future.

CAT activity was enhanced in Co-exposed algae, except for the highest CoCl_2_ concentration applied (Figure 5f). It could be speculated that in cells exposed to such severe stress, CAT molecules were damaged. It is known that CAT may be inhibited by peroxyradicals [77], and we observed an increase in lipid peroxidation in series with the high dose of CoCl_2_ in our model.

APX activity decreased for lower CoCl_2_ concentrations applied and increased for the highest (Figure 5e). APX is a heme-containing enzyme targeted to chloroplasts and mitochondria in *C. reinhardtii* [3,78]. As Co inhibits enzymes that play a role in the synthesis of a common precursor of heme and Chl [57], it is possible that under Co-induced stress there is competition between those pathways. As a result, less heme would be available in chloroplasts for the formation of APX holoenzymes, leading to a decrease in enzyme activity. The increase in prenyllipid antioxidants and Asc would compensate for this decrease, allowing efficient antioxidant protection. However, in a severely affected culture, in which the content of plastid prenyllipid antioxidants was decreased, the synthesis of APX could be prioritised, resulting in an increase in its activity. In Cd-exposed *C. reinhardtii* APX activity increased in a gradual dose-dependent manner, but much more pronounced increase occurred for the highest concentration of CdCl_2_ applied in the presence of an inhibitor of α-Toc and PQH_2_ synthesis [43]. Such an effect leads us to hypothesise that it may be a threshold-based response related to changes in cellular signalling and gene expression. This topic needs to be studied in more details in the future.

Hydrophilic low-molecular-weight antioxidants are important for protection against HM-induced stress. The amount of soluble thiols increased in Hg-, Cr-, Cd-, and Ni-exposed *C. reinhardtii* [43,47,79], Cu-exposed *Scenedesmus bijugatus* [80], Cu-, Ni-, and Zn-exposed *S. capricornutum* [81], and Cr-exposed *Monoraphidium convolutum* [82]. HM treatment also led to an increase in Asc level, for example, in Cd-, Cr-, and Cu-exposed *C. reinhardtii* [43,44], as well as in Cu-, and Zn-exposed *Chlorella sorokiniana* and *Scenedesmus acuminatus* [73,83]. The Pro content increased in Hg-, Cd-, Cr-, and Ni-exposed *C. reinhardtii* [43,47,70], Cd-, and Cu-exposed *Chlorella* sp. [84], Cu-, Ni-, and Zn-exposed *C. vulgaris* [63,85], Cu-exposed *C. sorokiniana* and *S. acuminatus* [73], Zn-exposed *S. acuminatus* [83], Pb-exposed *Scenedesmus obliquus* [86], as well as in Cu-, and Zn-exposed *Scenedesmus* sp. [87]. To our knowledge, the content of low-molecular-weight antioxidants has not been a subject of intensive research in eukaryotic algae exposed to Co. We managed to find only one study, in which exposure to Co was shown to cause an increase in the GSH content in *P. viridis* [18]. In addition, due to redundancy in the antioxidant response to HM-induced stress, it is important to measure more compounds in one model, to evaluate their interactions [43].

In the present experiment, we observed a significant dose-dependent increase in soluble thiols and Asc, suggesting that these compounds play an important protective role in Co-induced stress in *C. reinhardtii* (Figure 5a,b). The coordinated increase in thiols and Asc is important, because, apart from their individual action in ROS scavenging, the major antioxidant low-molecular thiol, GSH, plays a crucial role in the regeneration of Asc [3]. On the other hand, exposure to Co did not cause an increase in Pro content (Figure 5c). This may be surprising considering the common role of its imino acid in the responses to HMs. However, the increase in Pro does not always occur in HM-treated algae. For example, in our previous experiments, exposure to Cu caused an increase in Pro level in *C. reinhardtii* cell wall-deficient strain CW15, but this effect did not occur in strain 11-32b containing cell wall [44,88]. A decrease in Pro content was observed in Cd-exposed *Scenedesmus* sp. IITRIND2. The authors noticed that the proline biosynthetic pathway shares a common precursor, glutamate, with the GSH and phytochelatin biosynthetic pathway. They hypothesised that under Cd stress *Scenedesmus* sp. IITRIND2 would prefer the formation of phytochelatins rather than proline [89]. In the present study, an increase in the thiol content was observed; therefore, this could be the possible explanation of our results.

Prenyllipid antioxidants, located in plastids, i.e., tocopherols and PQH_2_, turned out to be an important player in antioxidant protection in photosynthetic organisms [20]. There are studies on their participation in the response to HM-induced stress, but they are less numerous compared to studies on hydrophilic antioxidants [3]. An increase in α-Toc content was observed in Cu-, Cd-, and Cr-exposed *C. reinhardtii* [41,42,43], Cu-, and Zn-exposed *C. sorokiniana* and *S. acuminatus* [73,83], as well as in Cu-, and Cd-exposed *Arabidopsis thaliana* [90]. The level of PQH_2_ increased in *C. reinhardtii* in response to Cr, Cd, and Hg ions [42,43]. Furthermore, Cu-tolerant strains of *C. reinhardti*, obtained as a result of prolonged growth in media with increased Cu content, contained more α-Toc and PQH_2_ than non-tolerant parent strain [69]. In our experiment, a dose-dependent increase in the content of these prenyllipids was also observed, suggesting that these antioxidants are important for the protection of *C. reinhardtii* during Co-induced stress (Figure 6). The decrease in α-Toc and PQH_2_ for the highest CoCl_2_ concentrations tested shows the exhaustion of protective mechanisms and can be correlated with the progress of lipid peroxidation and severe inhibition of growth (Figure 1a and Figure 4b).

## 4. Materials and Methods

The *C. reinhardtii* strain used in the present study was 11-32b (SAG collection, Göttingen, Germany). For all experiments, the algae were inoculated to provide an initial Chl *a* + *b* concentration of 0.5 μg/mL, and were grown for 2 weeks under continuous white light (50 μmol photons m^−2^ s^−1^), at 22 °C, on the shaker, as described in [88]. Modified Sager-Granick medium (3.75 mM NH_4_NO_3_, 1.22 mM MgSO_4_·7H_2_O, 0.73 mM KH_2_PO_4_, 0.57 mM K_2_HPO_4_, 0.36 mM CaCl_2_·2H_2_O, 37 μM FeCl_3_, 16.2 μM H_3_BO_4_, 0.84 μM CoCl_2_·6H_2_O, 0.24 μM CuSO_4_·5H_2_O, 2.02 μM MnCl_2_·4H_2_O, 0.83 μM (NH_4_)_6_Mo_7_O_24_·4H_2_O, 3.5 μM ZnSO_4_·7H_2_O, 5 mM HEPES pH 6.8 [42]) was used in the control series and as a basis for the media containing excessive Co. The latter were obtained by adding certain volumes of 50 mM CoCl_2_·6H_2_O. The basic salts and buffer for the media preparation were bought in Avantor Performance Materials Poland S.A. (Gliwice, Poland) and Merck (Darmstadt, Germany).

The applied CoCl_2_ concentrations were selected on the basis of preliminary experiments, where a wide range of concentrations were tested. In these experiments, we monitored algal growth for 2 weeks. The results of the growth rate measurements are shown in the Appendix A. We noticed that the application of CoCl_2_ in the range 2.5–20 μM did not cause significant growth inhibition. The application of 25 and 40 μM caused slight inhibition of growth. In the series containing 50 μM CoCl_2_ the growth was about 27% slower compared to the control. Growth inhibition of 50% occurred for 80 μM CoCl_2_. For CoCl_2_ concentrations ranging from 120 to 150 μM culture growth was about 85% slower than control and the amount of Chl in algal culture decreased compared to the initial value (Appendix A). In the experiment, in which culture growth, photosynthetic pigment content, and chlorophyll fluorescence parameters were evaluated, we decided to apply a wide range of CoCl_2_ concentrations: 2.5, 5, 15, 25, 40, 50, 80, and 125 µM. Among the series mentioned above, cultures containing 2.5, 15, 25, 40, and 80 µM CoCl_2_ were also used to determine markers of oxidative stress. As we did not observe an increase in oxidative stress markers in the tested range (except for the LOOH content after 2 weeks in the series with 80 µM CoCl_2_), we decided not to probe the response for more concentrations. All the above-mentioned measurements were carried out after 1 and 2 weeks of culture growth. The algae were grown in triplicates. For the determination of OD, one sample per each culture flask was taken. For measurements of photosynthetic pigments and Chl fluorescence, as well as for the assessment of oxidative stress markers, two samples per flask were taken.

Measurements of the antioxidant response required relatively large volumes of cultures to be taken per sample; therefore, we limited the CoCl_2_ concentrations applied for 15 and 30 μM (expected none and slight effect of algal growth, respectively), 50 and 80 μM (expected significant growth inhibition), and 120 µM (expected maximal growth inhibition for a tested range). The samples were collected after 2 weeks of growth. This sampling time was chosen because we wanted to compare the results obtained with the results of our previous studies on the response of *C. reinhardtii* to heavy metal ions [42,43]. Additionally, based on our previous experiments, it was known that the antioxidant response in *C. reinhardtii* was pronounced at the time point in which the difference in growth and Chl content between the HM-treated and control series was the most pronounced, and the markers of oxidative stress were enhanced in stressed cultures compared to the control [44]. In Co-treated *C. reinhardtii*, the effects related to toxicity were more pronounced after 2 weeks than after 1 week, which was another reason for choosing this time point. Algae were grown in triplicate, and for each type of measurement, one sample was taken per each flask.

Algal growth was monitored as OD at λ = 750 nm. Photosynthetic pigments were extracted with acetone as described in [43] and determined spectrophotometrically according to Lichtenthaler [91].

Chlorophyll fluorescence parameters, that is, maximum quantum yield of PSII photochemistry (F_v_/F_m_), and non-photochemical quenching (NPQ), calculated as (F_m_ − F_m_’)/F_m_’ (where F_m_ is maximum fluorescence in the dark-adapted state, while F_m_’ is maximum fluorescence in algae exposed to actinic light), were measured using Open FluorCam FC 800-O (Photon Systems Instruments, Brno, Czech Republic), as described in [42,43]. Before measurements, the algal suspensions were thickened to provide total Chl concentrations 5 μg/mL and portioned into 48-well plate. The algae were then dark-adapted for 30 min, pre-illuminated with very weak red light < 4 µM photons m^−2^ s^−1^ for 10 min to provide the photosynthetic apparatus being in state 1 [27], and the F_v_/F_m_ parameter was measured. The induction of NPQ was measured during 30 min of illumination with red actinic light of intensity of 200 μmol photons m^−2^ s^−1^; then, the actinic light was turned off, and the measurements were continued for 25 min. The saturating pulses (white light of intensity 2700 μmol photons m^−2^ s^−1^) were applied as indicated in the figures, to allow measuring of F_m’_.

Lipid hydroperoxides were measured using the fluorescent probe Spy-LHP, 2-(4-diphenylphosphanyl-phenyl)-9-(1-hexyl-heptyl)-anthra(2,1,9-def,6,5,10-d’e’f’)-diisoquinoline-1,3,8,10-tetraone (Dojindo, Kumamoto, Japan) as described in [42]. Briefly, the acetone stock solution of the probe was added to ethanol extract obtained of algal pellets to provide the final concentration of the probe, 10 µM; the samples were then incubated for 10 min in the dark, centrifuged (5 min, 9000× *g*), and the fluorescence spectrum was measured at λ _ex_ = 465 nm, with an emission range 500–600 nm. Cumulative O_2_^−•^ production in *C. reinhardtii* cells was analysed using in vivo nitrotetrazolium blue (NBT, Thermo Fisher Scientific, Waltham, Massachusetts, USA) staining as described in [88].

The selected hydrophilic antioxidants, i.e., Asc, soluble thiols, and Pro, as well as the activity of selected ROS-detoxifying enzymes, i.e., SOD, CAT, and APX, were measured as described in [88]. The reagents for the measurements, except for NBT mentioned above, were bought in Merck (Darmstadt, Germany). There were some minor modifications during sample preparation: 40 mL of algal culture was taken per sample instead of 30 mL; the algal pellets were rinsed with 50 mM potassium phosphate buffer pH 7.0 instead of growth medium; because strain 11-32b used in the present study has a cell wall, the sonication parameters were different (35% amplitude, 90 s total time of ultrasound emission in an on/off cycle of 9 s on/27 s off).

Hydrophobic antioxidants, i.e., α-Toc, and PQH_2_, were extracted with methanol and measured using the RP-HPLC system Jasco LC-4000 (Jasco Corporation, Tokyo, Japan) as described in [43], with minor modifications regarding solvent proportions. Briefly, the column used was C_18_ Tracer Excel 120 ODS-A (5 μm, 25 cm × 0.4 cm) and separation conditions were as follows: methanol/*n*-hexane (340:60, *v*/*v*), flow rate 1.6 mL min^−1^, absorbance detection at λ = 255 nm, and fluorescence detection at λ_ex_ = 290 nm, λ_em_ = 330 nm.

Data statistical analysis was carried out using STATISTICA 13.3. The following analyses have been performed: one-way ANOVA and post hoc Tukey’s test to compare means.

## 5. Conclusions

The examined CoCl_2_ concentration range allowed one to observe the typical response to HM toxicity, that is, tolerance for the lower concentrations applied and the exhaustion of protective mechanisms for the higher ones. In the first case, *C. reinhardtii* was able to efficiently activate the antioxidant response and protect itself from oxidative damage. Among the low-molecular-weight antioxidants which content increased in Co-exposed cells, there were important hydrophilic antioxidants (soluble thiols, Asc) and lipophilic antioxidants (α-Toc, PQH_2_). Among ROS-detoxifying enzymes, an increase in CAT activity was observed. Exhaustion of protective mechanisms in series exposed to high concentrations of CoCl_2_ resulted in significant growth inhibition, damage to the photosynthetic apparatus (decreased Chl content, Chl *a* to Chl *b* ratio, maximum quantum yield of PS II, and efficiency of NPQ induction), and increased lipid peroxidation. Furthermore, high concentrations of CoCl_2_ were able to inhibit dark-induced state transitions in *C. reinhardtii*, which has not been described before in the literature.

## Figures and Tables

**Figure 1 plants-14-03496-f001:**
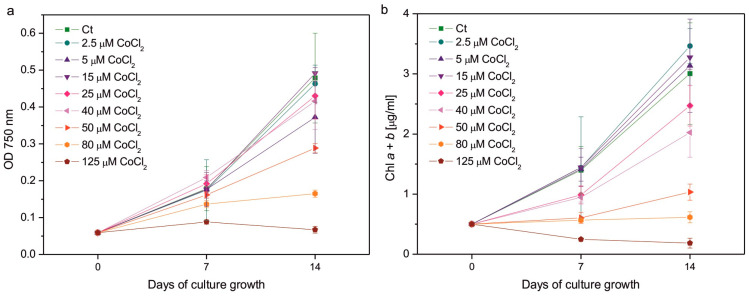
Culture growth followed by optical density at 750 nm (**a**), and chlorophyll *a* + *b* content (**b**) measured on the 7th and 14th days of growth of Co-exposed *C. reinhardtii*. The applied CoCl_2_ concentrations are given in the legends. Data are means ± SD (*n* = 3 for OD measurements, *n* = 6 for Chl measurements). Chl, chlorophyll; Ct, control; OD, optical density.

**Figure 2 plants-14-03496-f002:**
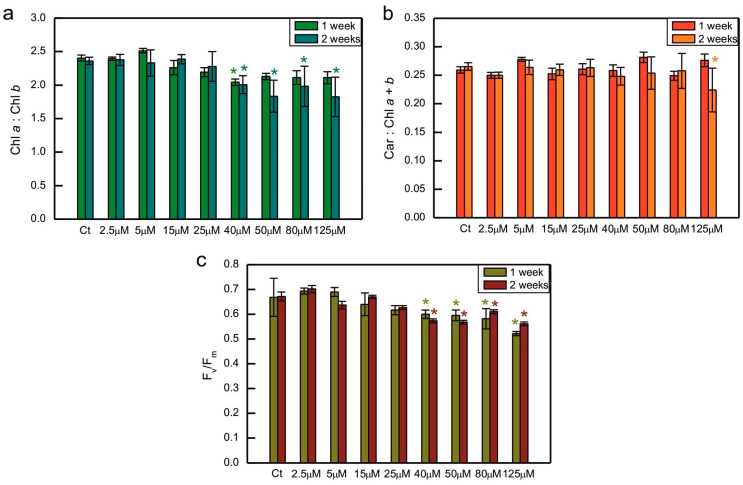
Chlorophyll *a* to chlorophyll *b* ratio (**a**), the ratio of total carotenoids to total chlorophyll (**b**), and the F_v_/F_m_ parameter representing the maximum quantum yield of photosystem II (**c**) in *C. reinhardtii* exposed to concentrations of CoCl_2_ indicated in the plots for 2 weeks. Photosynthetic pigment ratios are expressed as the ratios of pigment content expressed in [µg/mL of algal culture]. Data are means ± SD (*n* = 6). All means were compared using post hoc Tukey’s test, but for better clarity, only statistically significant differences between Co-exposed algae and the respective controls were marked with asterisks, * *p* < 0.05. Car, total carotenoids; Chl *a*, chlorophyll *a*; Chl *b*, chlorophyll *b*; Ct, control; F_m_, maximum fluorescence yield in the dark-adapted state; F_v_, variable fluorescence, that is, the difference between F_m_ and minimum fluorescence in the dark-adapted state, F_0_.

**Figure 3 plants-14-03496-f003:**
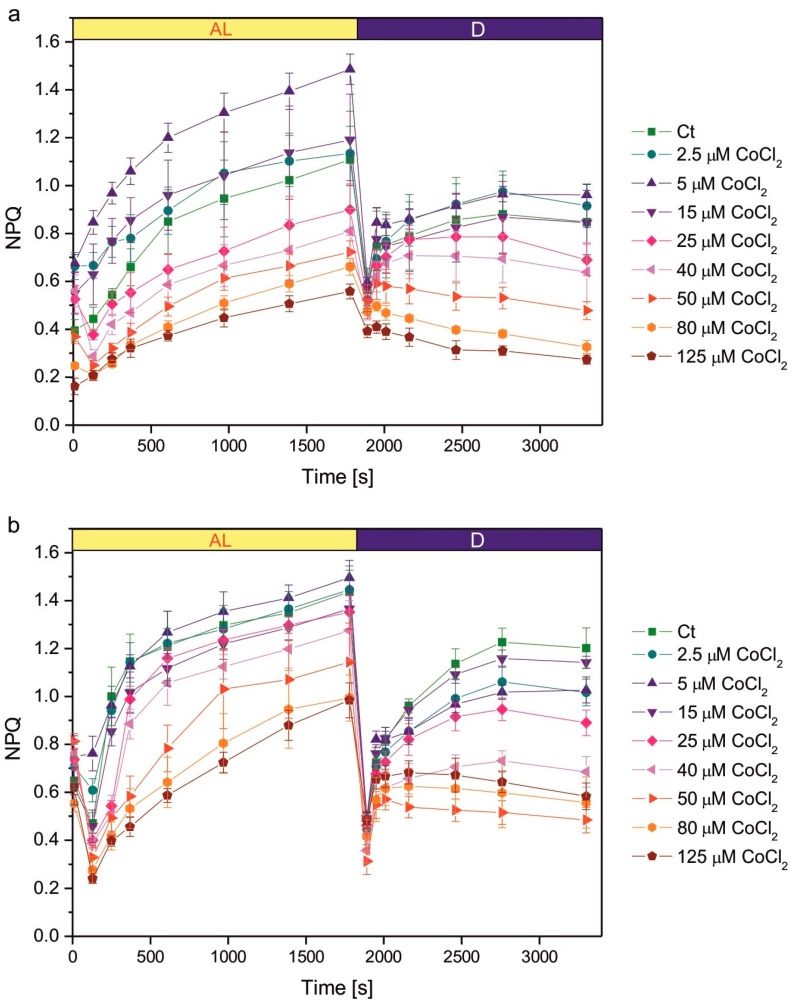
Induction of NPQ in 1-week-old (**a**) and 2-week-old (**b**) cultures of Co-exposed *C. reinhardtii*. The applied CoCl_2_ concentrations are given in the legend. The algae were exposed to red actinic light of intensity 200 μmol photons m^−2^ s^−1^ for 1800 s, then kept in the darkness for the next 1500 s. The timing of the saturating pulses is indicated in the plots. Data are means ± SD (*n* = 6). AL, actinic light; Ct, control; D, darkness; NPQ, parameter representing non-photochemical quenching of chlorophyll fluorescence.

**Figure 4 plants-14-03496-f004:**
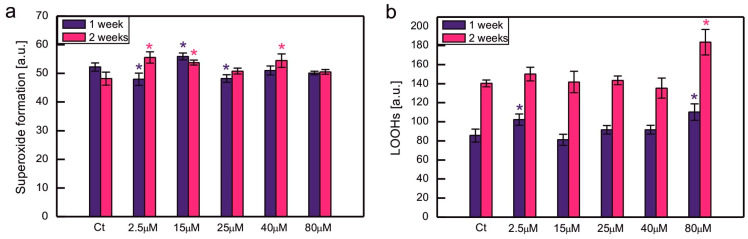
Superoxide formation measured semi-quantitatively as colour intensity after NBT staining (**a**) and lipid hydroperoxides monitored using the fluorescent probe Spy-LHC (**b**) in *C. reinhardtii* exposed to concentrations of CoCl_2_ indicated in the plots for 2 weeks. Data are means ± SD (*n* = 6). All means were compared using post hoc Tukey’s test, but for better clarity, only statistically significant differences between Co-exposed algae and respective controls were marked with asterisks, * *p* < 0.05. Ct, control; LOOHs, lipid hydroperoxides.

**Figure 5 plants-14-03496-f005:**
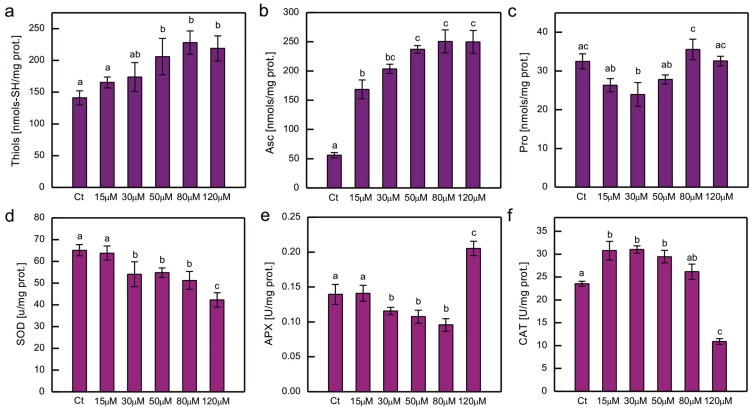
The content of hydrophilic antioxidants, that is, total soluble thiols (**a**), ascorbate (**b**), and proline (**c**), and the activity of ROS-detoxifying enzymes, that is, superoxide dismutase (**d**), ascorbate peroxidase (**e**), and catalase (**f**), in *C. reinhardtii* exposed to concentrations of CoCl_2_ indicated in the plots for 2 weeks. Data are means ± SD (*n* = 3). Different letters denote means that differ from each other with statistical significance *p* < 0.05 (post hoc Tukey’s test). APX, ascorbate peroxidase; Asc, ascorbate in its reduced form; CAT, catalase; Ct, control; Pro, proline; SOD, superoxide dismutase.

**Figure 6 plants-14-03496-f006:**
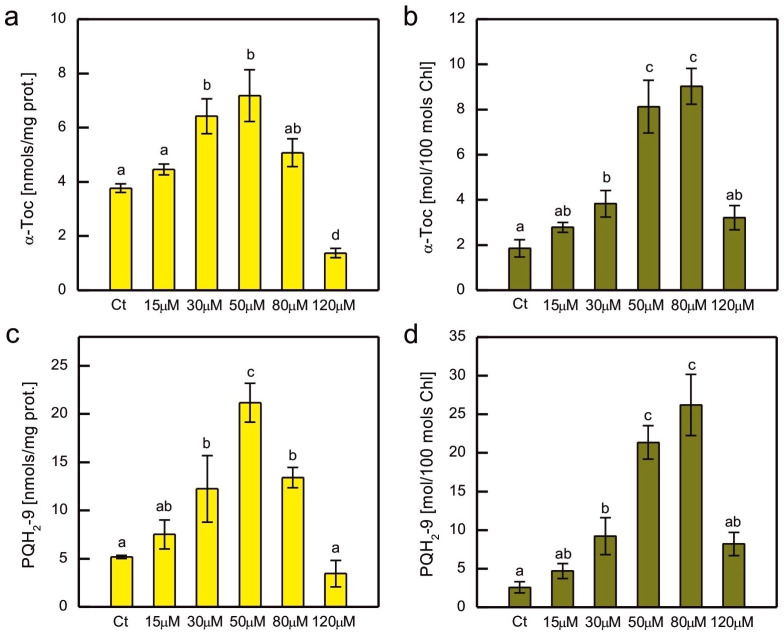
The content of lipophilic antioxidants, i.e., α-tocopherol (**a**,**b**) and plastoquinol (**c**,**d**), normalised to the content of soluble protein (**a**,**c**) or the content of chlorophyll *a* + *b* (**b**,**d**) in *C. reinhardtii* exposed to concentrations of CoCl_2_ indicated in the plots for 2 weeks. Data are means ± SD (*n* = 3). Different letters denote means that differ from each other with statistical significance *p* < 0.05 (post hoc Tukey’s test). α-Toc, α-tocopherol; Ct, control; PQH_2_, plastoquinol.

## Data Availability

The original contributions presented in the study are included in the article material, as well as in the Appendix A. Further inquiries can be directed to the corresponding author.

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
