# Peer review of "Analysis of the Response of Chlamydomonas reinhardtii to Cobalt Ions Reveals the Protective Role of Thiols, Ascorbate, and Prenyllipid Antioxidants, and the Negative Impact of Cobalt Toxicity on Photoprotective Mechanisms"

_plants, 2025, doi:10.3390/plants14223496_

Round 1

Reviewer 1 Report

Comments and Suggestions for Authors

In this study, the authors assessed the impact of CoCl2 on growth, photosynthetic pigment content, maximum quantum yield of photo-system II, efficiency of nonphotochemical quenching of chlorophyll fluorescence, oxidative stress markers, and antioxidant defence mechanisms in Chlamydomonas reinhardtii. Overall, the manuscript is well written and of novelty. I think it is suitable for publication in Plants. My main comments are as follow:

  1. Abstract should be enriched via specific valuable data which pave the way for understanding the study.
  2. The effects of heavy metals on plant growth have been widely reported (e.g. Total Environ. 2024, 957: 177858; 2023, 903: 166264). I suggest strength the Introduction and highlight the novelty of this study.
  3. Lines 400-401: What is the basis for the treatment levels?

Author Response

Reviewer 1

Respected Reviewer,

We are grateful for all your comments and suggestions.

1. Abstract should be enriched via specific valuable data which pave the way for understanding the study.

The abstract was extended.

2. The effects of heavy metals on plant growth have been widely reported (e.g. Total Environ. 2024, 957: 177858; 2023, 903: 166264). I suggest strength the Introduction and highlight the novelty of this study.

The introduction has been extended to include suggested references and to emphasize the novelty of the study.

3. Lines 400-401: What is the basis for the treatment levels?

Thank you for mentioning it. The detailed explanation was added to the materials and methods section, together with the supplementary file with the results of preliminary experiments.

Reviewer 2 Report

Comments and Suggestions for Authors

This paper describes a classic experiment on the study of cobalt toxicity in Chlamydomonas reinhardtii microalgae. The paper is well written and presents some interesting results but has however some flaws in the experimental part, hence I’m very sorry but my my recommendation is to reject:

In the first paragraph of the Materials and Methods section the authors indicate that apparently they used the following concentrations of Co:

2.5, 5, 15, 400 25, 40, 50, 80, and 125 μM

Of these, only 2.5, 15, 25, 40, and 80 were used for “markers of oxidative stress”. However afterwards the authors say that the concentrations of 15, 30, 50, 80, 120 μM were used for antioxidant response. Why were different concentrations used for different measurements? No explanation is given. This makes the analysis of the results much more difficult and is not a good experimental practice.

Also, I consider almost mandatory in this type of experiments on the toxicity of heavy metals, that the cobalt content be quantified in the algae, and unfortunately this was not done in the current work.

Of course, I would be happy to revise my decision if these issues were corrected.

Author Response

Respected Reviewer,

Please, find our answers below.

1. In the first paragraph of the Materials and Methods section the authors indicate that apparently they used the following concentrations (…) Why were different concentrations used for different measurements? No explanation is given. This makes the analysis of the results much more difficult and is not a good experimental practice.

The detailed explanation was added to the materials and methods section, together with the supplementary file with the results of preliminary experiments. I agree that for the clarity of the comparison of the results of NPQ measurements and antioxidant determination, the 125 mM CoCl2 should be applied in the experiment with antioxidant determination, not 120 mM CoCl2. It was my mistake. However, both of these concentrations were in the range when the growth was nearly totally inhibited and this 5 mM difference did not cause difference in overall response. Actually, at first I rather wanted use series with 120 mM as sort of positive control for high toxicity, and I did not plan to show the results from this series (I was afraid that the signals would be too weak to give satisfactory signal to noise ratio). However, as these results turned to be interesting, I decided to report them too.

2. Also, I consider almost mandatory in this type of experiments on the toxicity of heavy metals, that the cobalt content be quantified in the algae, and unfortunately this was not done in the current work.

The information concerning quantification of the metal in the algae definitely adds to the overall results.

However, the aim of the present study was to observe various aspects of physiological and biochemical responses to Co-inducted stress and we were interested in relating these responses to the effect on growth, which, in our opinion, reflects general toxicity.

Simple measurements of metal content in biomass are very good for the experiments concerning the potential of given species for phytoremediation as they reflect general biding of metal ions by cells.

Although, it is known that the majority of heavy metal ions is bound in algal cell wall.

Wang, L., Liu, J., Filipiak, M., Mungunkhuyag, K., Jedynak, P., Burczyk, J., ... & Malec, P. (2021). Fast and efficient cadmium biosorption by Chlorella vulgaris K-01 strain: The role of cell walls in metal sequestration. Algal Research60, 102497.

The experiments with two C. reinhardtii strains a cell wall-containing one and a wall-less one, exposed to Cd, Cu, Co, and Ni, showed that about half of the Co contained by these algae is loosely bound to the cell wall.

Macfie, S. M., & Welbourn, P. M. (2000). The cell wall as a barrier to uptake of metal ions in the unicellular green alga Chlamydomonas reinhardtii (Chlorophyceae). Archives of Environmental Contamination and Toxicology, 39(4), 413-419.

Such an extracellular Co would be seen in the analysis, in which the Co content is measured in algal biomass, but it would not have an impact on the metabolism.

What is more, even the intracellular pool of heavy metal ions is not necessary related to toxic effect, as some of these ions may be sequestered in the vacuoles or as inclusions in various cell compartments.

Andosch, A., Höftberger, M., Lütz, C., & Lütz-Meindl, U. (2015). Subcellular sequestration and impact of heavy metals on the ultrastructure and physiology of the multicellular freshwater alga Desmidium swartzii. International Journal of Molecular Sciences, 16(5), 10389-10410.

The occurrence of various effects related to heavy metal toxicity in exposed algae, in our study, suggest that cobalt ions entered the cells.

The detailed evaluation of the heavy metal ions content in various cell compartments, enabling the assessment of these ions in metabolically active compartments is very advanced study, which could be performed for the sake of separate paper, dedicated only to intracellular deposition of heavy metal ions, such as the one cited below:

Vázquez-Arias, A., Rodríguez-Prieto, C., Yamada, Y., Ito, M., Fernández, J. Á., & Aboal, J. R. (2025). Deciphering uptake mechanisms of potentially toxic elements in seaweeds using high resolution imaging analysis. Journal of Hazardous Materials, 139646.

In our paper, we rather wanted to carry out analysis comparable to those present – for example – in:

Elbaz, A., Wei, Y. Y., Meng, Q., Zheng, Q., & Yang, Z. M. (2010). Mercury-induced oxidative stress and impact on antioxidant enzymes in Chlamydomonas reinhardtii. Ecotoxicology, 19(7), 1285-1293.

Luis, P., Behnke, K., Toepel, J., & Wilhelm, C. (2006). Parallel analysis of transcript levels and physiological key parameters allows the identification of stress phase gene markers in Chlamydomonas reinhardtii under copper excess. Plant, Cell & Environment29(11), 2043-2054.

Li, M., Hu, C., Zhu, Q., Chen, L., Kong, Z., & Liu, Z. (2006). Copper and zinc induction of lipid peroxidation and effects on antioxidant enzyme activities in the microalga Pavlova viridis (Prymnesiophyceae). Chemosphere, 62(4), 565-572.

And to our former studies concerning antioxidant response in C. reinhardtii under HM-induced stress:

Nowicka, B., Pluciński, B., Kuczyńska, P., & Kruk, J. (2016). Physiological characterization of Chlamydomonas reinhardtii acclimated to chronic stress induced by Ag, Cd, Cr, Cu and Hg ions. Ecotoxicology and Environmental safety, 130, 133-145.

Nowicka, B., Zyzik, M., Kapsiak, M., Ogrodzińska, W., & Kruk, J. (2021). Oxidative stress limits growth of Chlamydomonas reinhardtii (Chlorophyta, Chlamydomonadales) exposed to copper ions at the early stage of culture growth. Phycologia, 60(4), 303-313.

Nowicka, B., Fesenko, T., Walczak, J., & Kruk, J. (2020). The inhibitor-evoked shortage of tocopherol and plastoquinol is compensated by other antioxidant mechanisms in Chlamydomonas reinhardtii exposed to toxic concentrations of cadmium and chromium ions. Ecotoxicology and Environmental Safety, 191, 110241.

Where heavy metal ions content in algae was not measured.

Reviewer 3 Report

Comments and Suggestions for Authors

The Ms titled: The antioxidant response plays a role in the acclimation of 2 Chlamydomonas reinhardtii to stress induced by cobalt ions submitted by Aylin Köktena and Beatrycze Nowicka from Jagiellonian University is very interesting. But some improvements are necessary.

(1) Title should be improved

(2) In introduction part should focus on chemical behavior, bioavailability, and toxicity rather than density alone. Improve the introduction part.

(3) Material and Methods Section: It is said that cobalt-free medium was as a negative control for all experiments. Triplicates are mentioned for cultures, but not for all downstream assays (e.g., lipid hydroperoxide measurements). Were all experiments performed in triplicate?

(4) Preliminary experiments: The text says concentrations were chosen “on the basis of preliminary experiments,” but details are missing (e.g., what range caused complete inhibition vs. tolerance). Add a brief summary or a citation to supplementary data.

(5) Results and discussion: The discussion mostly reiterates results (growth and chlorophyll inhibition under Co exposure) without delving into mechanistic explanations. For example: How does cobalt specifically interfere with chlorophyll biosynthesis compared to cadmium or copper?

(6) Are differences between species due to metal uptake, detoxification mechanisms, or pigment composition? Add clear explanation.

(7)  The figure 3 is "not clear," improve it

(8) The conclusion should not be a repetition of the result; it needs to be rewritten

Author Response

Respected Reviewer,

We are grateful for all your valuable comments and suggestions. Please, find a response to them below.

(1) Title should be improved

The title has been changed to better reflect the major observations.

(2) In introduction part should focus on chemical behavior, bioavailability, and toxicity rather than density alone. Improve the introduction part.

The introduction has been extended accordingly.

(3) Material and Methods Section: It is said that cobalt-free medium was as a negative control for all experiments. Triplicates are mentioned for cultures, but not for all downstream assays (e.g., lipid hydroperoxide measurements). Were all experiments performed in triplicate?

Thank you for noticing this. In some experiments we did two technical repetitions per culture. We extended materials and methods section and added this information.

(4) Preliminary experiments: The text says concentrations were chosen “on the basis of preliminary experiments,” but details are missing (e.g., what range caused complete inhibition vs. tolerance). Add a brief summary or a citation to supplementary data.

Thank you for this comment. The detailed explanation was added to the materials and methods section, together with the supplementary file with the results of preliminary experiments.

(5) Results and discussion: The discussion mostly reiterates results (growth and chlorophyll inhibition under Co exposure) without delving into mechanistic explanations. For example: How does cobalt specifically interfere with chlorophyll biosynthesis compared to cadmium or copper?

The discussion has been extended. The information concerning of the impact of Co on Chl biosynthesis was added, however we did not delve much into comparison with other HMs. Many of them act at similar biosynthetic steps, there are also differences in the action of certain metal at specific steps, but detailed description of these would be quite long. We quoted the review paper concerning the impact of various HMs on Chl biosynthesis.

(6) Are differences between species due to metal uptake, detoxification mechanisms, or pigment composition? Add clear explanation.

The explanation was added.

(7)  The figure 3 is "not clear," improve it

The figure was changed, we hope that it will make it more clear.

(8) The conclusion should not be a repetition of the result; it needs to be rewritten

The conclusions were rewritten.

Reviewer 4 Report

Comments and Suggestions for Authors

This manuscript comprehensively investigates the physiological and antioxidant responses of the green alga Chlamydomonas reinhardtii to cobalt (Co) stress. The first key finding is that at lower, sub-lethal Co concentrations, the alga activates a multifaceted antioxidant defense system (e.g., increased thiols, ascorbate, α-tocopherol, plastoquinol, and catalase activity) which effectively prevents oxidative damage. The second major point is that higher, toxic Co concentrations overwhelm this protective capacity, leading to significant growth inhibition, photosynthetic damage (reduced Fv/Fm, NPQ, Chl content), and the onset of oxidative stress (increased lipid peroxidation). It is a manuscript with a comprehensive and interesting experimental plan. If the author can make revisions or consider the following details, it is a highly recommended manuscript for readers.

  1. Inconsistent Co concentration ranges: The concentrations used in the growth/pigment/fluorescence experiments (up to 125 µM) differ from those in the antioxidant assays (up to 120 µM). Why was this done? It complicates direct correlation of effects across the full dose-response spectrum.
  2. Limited oxidative stress markers: The assessment of oxidative stress relies heavily on lipid hydroperoxides (LOOH) and a semi-quantitative NBT stain for O₂•⁻. If hydrogen peroxide can be further quantified or more specific/sensitive methods (e.g., DCFH-DA) can be used, it will be beneficial to strengthen the evidence of no oxidative stress at low cobalt doses, or it would be more meaningful to mention it in the outlook.
  3. Ambiguity in SOD activity results: The consistent dose-dependent decrease in SOD activity is intriguing but not sufficiently discussed. Is this due to direct inhibition by Co²⁺, downregulation of expression, or enzyme degradation? Measuring mRNA levels or using native PAGE activity staining could provide mechanistic insight.
  4. Lack of internal Co quantification: The actual intracellular Co accumulation levels are not measured. This data is crucial to link the external concentration to the observed physiological effects and to compare toxicity with other heavy metals.
  5. Overinterpretation of the "acclimation" concept: The title and conclusion suggest "acclimation," but the data primarily shows a stress response. To robustly claim acclimation, evidence of recovered growth or physiological parameters over a longer period in sustained low Co stress would be needed.
  6. Incomplete discussion of APX response: The decrease in APX activity at medium Co doses followed by an increase at the highest dose is noted but the biological rationale is speculative. Could this be related to differential compartmentalization (e.g., plastid vs. mitochondrial APX isoforms) or a threshold-based stress response?
  7. Justification for time points: The rationale for measuring antioxidants only at the 2-week endpoint is not provided. A time-course study (e.g., early vs. late response) could reveal the dynamics of the antioxidant induction.
  8. Clarity on NPQ in darkness: The observation that high Co affects NPQ in darkness (suggesting impact on chlororespiration/state transitions) is novel but preliminary. The methodology description for this specific measurement could be clearer. How was the "darkness" value quantified from the trace?
  9. Contextualization with existing literature: The discussion is thorough but could better highlight what is genuinely novel here compared to previous studies on Co or other heavy metals in *C. reinhardtii* (e.g., the specific pattern of lipophilic antioxidant response).

Author Response

Respected reviewer,

We are grateful for in depth analysis of our paper. Please, find your questions and suggestions addressed below.

1. Inconsistent Co concentration ranges: The concentrations used in the growth/pigment/fluorescence experiments (up to 125 µM) differ from those in the antioxidant assays (up to 120 µM). Why was this done? It complicates direct correlation of effects across the full dose-response spectrum.

The detailed explanation was added to the materials and methods section, together with the supplementary file with the results of preliminary experiments. I agree that for the clarity of the comparison of the results of NPQ measurements and antioxidant determination, the 125 mM CoCl2 should be applied in the experiment with antioxidant determination, not 120 mM CoCl2. It was my mistake, for which I am deeply sorry. However, both of these concentrations were in the range when the growth was nearly totally inhibited and this 5 mM difference did not cause difference in overall response. Actually, at first I rather wanted use series with 120 mM as sort of positive control for high toxicity, and I did not plan to show the results from this series (I was afraid that the signals would be too weak to give satisfactory signal to noise ratio). However, as these results turned to be interesting, I decided to report them too.

2. Limited oxidative stress markers: The assessment of oxidative stress relies heavily on lipid hydroperoxides (LOOH) and a semi-quantitative NBT stain for O₂•⁻. If hydrogen peroxide can be further quantified or more specific/sensitive methods (e.g., DCFH-DA) can be used, it will be beneficial to strengthen the evidence of no oxidative stress at low cobalt doses, or it would be more meaningful to mention it in the outlook.

DCFH-DA assay is widely used for monitoring of ROS formation. When I started research concerning oxidative stress in C. reinhardtii exposed to heavy metals in 2013, we worked on optimalisation of the protocol for the sake of our experiments and later used it in some models. However, the results obtained using these method raised our doubts. For example, the results obtained using monitoring of lipid peroxidation or NBT staining correlated well with other results (for example with antioxidant depletion), whereas the results of DCFH-DA protocol did not. What is more, sometimes the results were strange, for example, stronger DCFH-DA signal in control than in the series where a compounds known to induce oxidative stress were added.

There is literature data concerning the problems with DCFH-DA assay. It was shown that DCFH-DA may undergo oxidation with heme, cytochrome c and hemoglobin. This oxidation is independent from ROS.

Ohashi, T., Mizutani, A., Murakami, A., Kojo, S., Ishii, T., & Taketani, S. (2002). Rapid oxidation of dichlorodihydrofluorescin with heme and hemoproteins: formation of the fluorescein is independent of the generation of reactive oxygen species. FEBS letters, 511(1-3), 21-27.

DCFH-DA may also undergo conversion to fluorescent products under light, which makes serious problem when we use this probe for the measurements of photosynthetic cells.

Afzal, M., Matsugo, S., Sasai, M., Xu, B., Aoyama, K., & Takeuchi, T. (2003). Method to overcome photoreaction, a serious drawback to the use of dichlorofluorescin in evaluation of reactive oxygen species. Biochemical and biophysical research communications, 304(4), 619-624.

This probe was shown not to be able to directly measure H2O2, and to enter self-propagating redox-cycling reactions.

Kalyanaraman, B., Darley-Usmar, V., Davies, K. J., Dennery, P. A., Forman, H. J., Grisham, M. B., ... & Ischiropoulos, H. (2012). Measuring reactive oxygen and nitrogen species with fluorescent probes: challenges and limitations. Free radical biology and medicine52(1), 1-6.

That is why we do not use DCFH-DA method anymore.

As could be seen in the review summarizing the experiments on heavy metals and oxidative stress in eukaryotic algae (Nowicka, B. (2022). Heavy metal–induced stress in eukaryotic algae—mechanisms of heavy metal toxicity and tolerance with particular emphasis on oxidative stress in exposed cells and the role of antioxidant response. Environmental Science and Pollution Research, 29(12), 16860-16911), usually one or two oxidative stress markers were measured in one study, therefore, we considered two methods as enough for our study.

3. Ambiguity in SOD activity results: The consistent dose-dependent decrease in SOD activity is intriguing but not sufficiently discussed. Is this due to direct inhibition by Co²⁺, downregulation of expression, or enzyme degradation? Measuring mRNA levels or using native PAGE activity staining could provide mechanistic insight.

The discussion section has been extended to provide plausible explanation.

4. Lack of internal Co quantification: The actual intracellular Co accumulation levels are not measured. This data is crucial to link the external concentration to the observed physiological effects and to compare toxicity with other heavy metals.

The information concerning quantification of the metal in the algae definitely adds to the overall results.

However, the aim of the present study was to observe various aspects of physiological and biochemical responses to Co-inducted stress and we were interested in relating these responses to the effect on growth, which, in our opinion, reflects general toxicity.

Simple measurements of metal content in biomass are very good for the experiments concerning the potential of given species for phytoremediation as they reflect general biding of metal ions by cells.

Although, it is known that the majority of heavy metal ions is bound in algal cell wall.

Wang, L., Liu, J., Filipiak, M., Mungunkhuyag, K., Jedynak, P., Burczyk, J., ... & Malec, P. (2021). Fast and efficient cadmium biosorption by Chlorella vulgaris K-01 strain: The role of cell walls in metal sequestration. Algal Research60, 102497.

The experiments with two C. reinhardtii strains a cell wall-containing one and a wall-less one, exposed to Cd, Cu, Co, and Ni, showed that about half of the Co contained by these algae is loosely bound to the cell wall.

Macfie, S. M., & Welbourn, P. M. (2000). The cell wall as a barrier to uptake of metal ions in the unicellular green alga Chlamydomonas reinhardtii (Chlorophyceae). Archives of Environmental Contamination and Toxicology, 39(4), 413-419.

Such an extracellular Co would be seen in the analysis, in which the Co content is measured in algal biomass, but it would not have an impact on the metabolism.

What is more, even the intracellular pool of heavy metal ions is not necessary related to toxic effect, as some of these ions may be sequestered in the vacuoles or as inclusions in various cell compartments.

Andosch, A., Höftberger, M., Lütz, C., & Lütz-Meindl, U. (2015). Subcellular sequestration and impact of heavy metals on the ultrastructure and physiology of the multicellular freshwater alga Desmidium swartzii. International Journal of Molecular Sciences, 16(5), 10389-10410.

The occurrence of various effects related to heavy metal toxicity in exposed algae, in our study, suggest that cobalt ions entered the cells.

The detailed evaluation of the heavy metal ions content in various cell compartments, enabling the assessment of these ions in metabolically active compartments is very advanced study, which could be performed for the sake of separate paper, dedicated only to intracellular deposition of heavy metal ions, such as the one cited below:

Vázquez-Arias, A., Rodríguez-Prieto, C., Yamada, Y., Ito, M., Fernández, J. Á., & Aboal, J. R. (2025). Deciphering uptake mechanisms of potentially toxic elements in seaweeds using high resolution imaging analysis. Journal of Hazardous Materials, 139646.

The determination of the effective concentration of heavy metal ions in metabolically active cell compartments is an interesting issue to be answered to compare toxicity of various heavy metals. However, in the studies concerning the determination of wider array of antioxidants and ROS-detoxifying enzymes in C. reinhardtii exposed to heavy metal-induced stress or in Co-exposed eukaryotic algae (see list below), such measurements have not been carried out yet. 

Elbaz, A., Wei, Y. Y., Meng, Q., Zheng, Q., & Yang, Z. M. (2010). Mercury-induced oxidative stress and impact on antioxidant enzymes in Chlamydomonas reinhardtii. Ecotoxicology, 19(7), 1285-1293.

Luis, P., Behnke, K., Toepel, J., & Wilhelm, C. (2006). Parallel analysis of transcript levels and physiological key parameters allows the identification of stress phase gene markers in Chlamydomonas reinhardtii under copper excess. Plant, Cell & Environment29(11), 2043-2054.

Li, M., Hu, C., Zhu, Q., Chen, L., Kong, Z., & Liu, Z. (2006). Copper and zinc induction of lipid peroxidation and effects on antioxidant enzyme activities in the microalga Pavlova viridis (Prymnesiophyceae). Chemosphere, 62(4), 565-572.

Nowicka, B., Pluciński, B., Kuczyńska, P., & Kruk, J. (2016). Physiological characterization of Chlamydomonas reinhardtii acclimated to chronic stress induced by Ag, Cd, Cr, Cu and Hg ions. Ecotoxicology and Environmental safety, 130, 133-145.

Nowicka, B., Zyzik, M., Kapsiak, M., Ogrodzińska, W., & Kruk, J. (2021). Oxidative stress limits growth of Chlamydomonas reinhardtii (Chlorophyta, Chlamydomonadales) exposed to copper ions at the early stage of culture growth. Phycologia, 60(4), 303-313.

Nowicka, B., Fesenko, T., Walczak, J., & Kruk, J. (2020). The inhibitor-evoked shortage of tocopherol and plastoquinol is compensated by other antioxidant mechanisms in Chlamydomonas reinhardtii exposed to toxic concentrations of cadmium and chromium ions. Ecotoxicology and Environmental Safety, 191, 110241.

5. Overinterpretation of the "acclimation" concept: The title and conclusion suggest "acclimation," but the data primarily shows a stress response. To robustly claim acclimation, evidence of recovered growth or physiological parameters over a longer period in sustained low Co stress would be needed.

Thank you for spotting this. We corrected it.

6. Incomplete discussion of APX response: The decrease in APX activity at medium Co doses followed by an increase at the highest dose is noted but the biological rationale is speculative. Could this be related to differential compartmentalization (e.g., plastid vs. mitochondrial APX isoforms) or a threshold-based stress response?

The discussion concerning APX has been extended.

7. Justification for time points: The rationale for measuring antioxidants only at the 2-week endpoint is not provided. A time-course study (e.g., early vs. late response) could reveal the dynamics of the antioxidant induction.

The explanation was added to the materials and methods section.

The present project is a continuation of the research on heavy metal toxicity in model microalgae C. reinhardtii carried out by our group. Therefore, we wanted our results to be comparable to the results of our former studies.

In the experiment, when the wider range of heavy metals were tested (but not as wide array of antioxidants was measured) we carried out the measurements in 2-week-old cultures.

Nowicka, B., Pluciński, B., Kuczyńska, P., & Kruk, J. (2016). Physiological characterization of Chlamydomonas reinhardtii acclimated to chronic stress induced by Ag, Cd, Cr, Cu and Hg ions. Ecotoxicology and Environmental safety, 130, 133-145.

In the case of Cd and Cr treatment the inhibitory impact of these ions on growth did not diminish in time. Later, we did more detailed analysis of antioxidant response in Cr- and Cd-treated algae, and we carried out measurements in 2-week-old cultures too.

Nowicka, B., Fesenko, T., Walczak, J., & Kruk, J. (2020). The inhibitor-evoked shortage of tocopherol and plastoquinol is compensated by other antioxidant mechanisms in Chlamydomonas reinhardtii exposed to toxic concentrations of cadmium and chromium ions. Ecotoxicology and Environmental Safety, 191, 110241.

In Cu-exposed algae, the toxic effect was more pronounced during early stage of growth. We did additional measurements with monitoring of antioxidant response at various time points and discovered that the antioxidant response was most pronounced at time point in which growth inhibition and oxidative stress markers were the most pronounced.

Nowicka, B., Zyzik, M., Kapsiak, M., Ogrodzińska, W., & Kruk, J. (2021). Oxidative stress limits growth of Chlamydomonas reinhardtii (Chlorophyta, Chlamydomonadales) exposed to copper ions at the early stage of culture growth. Phycologia, 60(4), 303-313.

When I took into account the kinetics of growth and Chl accumulation in cultures, response to Co was more similar to Cd and Cr. Plus, the toxic effect of Co (seen as growth inhibition, photosynthetic pigments decrease, maximum quantum yield of PS II decrease, and lipid peroxidation) was more pronounced after 2 weeks than after 1 week. Therefore I chose 2 weeks as a time point for sample collection and measurements.

In facts, I did experiments concerning early response to heavy metals (paper sent to journal, waiting for reviews). I measured lipophilic antioxidants. In these experiments it turned out that the concentrations known to cause severe growth inhibition in the situation when algae are grown in the presence of heavy metal salts in the medium from the beginning, did not have such effect if applied to the grown culture (this most probably results from binding of metal ions extracellularly). This makes the comparison of the two types of experimental setups difficult.

One could imagine experiment with the exposing algae to heavy metals at low cell density and taking samples early. But, for the measurements of antioxidants and enzymes we needed lot of biomass per sample, so the amount of cultures needed for such experiments would be large.

Plus, we wanted to measure NPQ in our algae, to compare results with our former experiments (Nowicka et al. 2016, see above). To do this, we needed to grow algae in the medium without high amount of acetate, because high amounts of acetate enhance growth of the algae significantly, but in such cultures the qE mechanism is diminished.

Nowicka, B. (2020). Practical aspects of the measurements of non‐photochemical chlorophyll fluorescence quenching in green microalgae Chlamydomonas reinhardtii using Open FluorCam. Physiologia plantarum, 168(3), 617-629.

However, a consequence of growing algae without high content of acetate leads to slow growth and low cell densities.

8. Clarity on NPQ in darkness: The observation that high Co affects NPQ in darkness (suggesting impact on chlororespiration/state transitions) is novel but preliminary. The methodology description for this specific measurement could be clearer. How was the "darkness" value quantified from the trace?

The description of the method has been extended. The “darkness” period means that actinic light is off during this time. However, both during actinic light exposure and the following darkness period the saturating flashes are applied. Saturating light is a flash of very strong light applied for 1 second to measure fluorescence of the PS II in its close state. This light induces strong fluorescence of the sample which is then measured as Fm’ parameter used to calculate NPQ parameter.

Also, more detailed explanation of the effects observed in the Figure 3 was added to the results section, to make the interpretation of the data more clear for general readers.

9. Contextualization with existing literature: The discussion is thorough but could better highlight what is genuinely novel here compared to previous studies on Co or other heavy metals in *C. reinhardtii* (e.g., the specific pattern of lipophilic antioxidant response).

Thank you for your suggestion. We extended the discussion, plus we added sentences to abstract and introduction, and we changed conclusions, to emphasize novelty of the study.

Reviewer 5 Report

Comments and Suggestions for Authors

The manuscript entitled “The antioxidant response plays a role in the acclimation of Chlamydomonas reinhardtii to stress induced by cobalt ions” has been submitted for possible publication in Plants-MDPI Journal. Authors addressed the effect of cobalt on growth, pigment content, photosynthetic efficiency, oxidative stress markers, and antioxidant responses in Chlamydomonas reinhardtii. The study is interesting and relevant, as cobalt toxicity is comparatively less explored than other heavy metals. The experimental design  is rational and manuscript is written well. However, there are some issues that need to be addressed before the manuscript can be considered for publication.

Major comments:

(1) In introduction part carefully review the literature and expand the literature review to include relevant studies on antioxidant responses in various algal species under heavy metal stress. This broader context will highlight the unique aspects of Chlamydomonas reinhardtii in coping with cobalt-induced stress.

(2) How you selected CoCl₂ concentrations (up to 125 µM). Were these concentrations chosen based on preliminary toxicity assays, and do they reflect natural exposure scenarios?

(3) The decrease in NPQ efficiency is an interesting finding. However, the interpretation that cobalt may inhibit chlororespiration activity is speculative. Could the authors provide stronger evidence, or otherwise phrase this more carefully as a hypothesis requiring future testing?

(3) The differential effects on SOD (decrease) vs CAT/APX (increase at certain concentrations) are interesting. The discussion should explain on possible mechanistic reasons why SOD is specifically inhibited by Co²⁺ in this model.

(4) For proline, the lack of accumulation is notable, given its common role in stress tolerance. The authors could expand on why C. reinhardtii may not use proline under cobalt stress.

Minor Comments

(5) Authors thoroughly revise the manuscript to avoid grammatical and typo errors

(6) Lines 31-33: The term ‘heavy metals’ (HMs) is often used for metals and metalloids whose density in the pure elemental state exceeds 5 g/cm³. Some of these elements are essential micronutrients, showing toxicity at concentrations exceeding the tolerance range of a given species, while the other do not play any positive role in biochemistry.

“the other do not” grammatical error; should be “others do not.”

(7) Lines 37-41: The most common mechanism of toxicity is due to the similarity of given HM ions to essential metal ions, resulting in the substitution of the latter in their binding sites in proteins and other compounds of biological importance, such as chlorophyll (Chl), as well as in competition for transport systems leading to a shortage of nutrients and a disturbance of ion and water homeostasis.

Sentence is too long and difficult to follow. Spilt it into simple sentences

Example: The most common mechanism of heavy metal toxicity is that these ions resemble essential metals. This similarity allows them to replace essential metals in proteins and important compounds such as chlorophyll, and to compete for transport systems, causing nutrient shortages and disrupting ion and water balance (Li et al., 2024; Yu et al., 2023). Add relevant reference here

References: Li, X., Xu, B., Sahito, Z. A., Chen, S., & Liang, Z. (2024). Transcriptome analysis reveals cadmium exposure enhanced the isoquinoline alkaloid biosynthesis and disease resistance in Coptis chinensis. Ecotoxicology and environmental safety271, 115940.

Yu, S., Sahito, Z. A., Lu, M., Huang, Q., Du, P., Chen, D., ... & Yang, X. (2023). Soil water stress alters differentially relative metabolic pathways affecting growth performance and metal uptake efficiency in a cadmium hyperaccumulator ecotype of Sedum alfredii. Environmental Science and Pollution Research30(38), 88986-88997.

(8) Line 43: These are mainly thioyl but also histidyl and carboxyl groups of proteins and low-molecular-weight compounds, such as glutathione (GSH).

Check the spelling “thioyl” or “thiol” is more correct in biochemical context.

Lines 61-62: Excess Co has also been shown to inhibit RNA synthesis and disturb mitotic spindle formation.

 “disturb” is not correct, use “disrupt.”

(9) Lines 88- 95 (Aim of study)

The aim of the present study was to evaluate the impact of toxic concentrations of CoCl2 on the growth, photosynthetic pigment content, selected photosynthesis-related parameters (maximum quantum yield of PSII and the efficiency of non-photochemical quenching of chlorophyll fluorescence), oxidative stress markers (O₂•– formation and the content of lipid hydroperoxides, LOOHs), hydrophilic antioxidants (soluble thiols, Asc, Pro), lipophilic antioxidants (α-Tocopherol, α-Toc, and plastoquinol, PQH₂), and ROS-detoxifying enzymes (SOD, CAT, APX), in model green microalgae C. reinhardtii, widely used in research on HMs toxicity and tolerance.

Very long sentence, difficult to explain. Revise it

(10) Lines 106- 107:...but the effect did not turn out to be statistically significant.

 “did not turn out to be.” is not clear. Improve it as:

Example: ...but this difference was not statistically significant.

(11) Lines 389–390:  For the sake of the experiments, algae were inoculated to provide an initial Chl a + b concentration of 0.5 µg/ml...

“For the sake of” is not clear.

Example: For all experiments, cultures were inoculated at an initial chlorophyll a + b concentration of 0.5 µg/ml.

(12) Line 429-31 ...and because the strain 11-32b used in the present study has cell wall, the sonication parameters were different...

Grammar: “has cell wall” should be “has a cell wall.”

(13) Lines 445–446; As it was expected, excessive Co caused inhibition of C. reinhardtii growth and decrease in total Chl content compared to the control.

“As it was expected” unnecessary; conclusion should be concise.

Author Response

Respected reviewer,

We are grateful for your taking time to read the manuscript carefully and suggest the improvements. We addressed your questions and suggestions below.

Major comments:

1. In introduction part carefully review the literature and expand the literature review to include relevant studies on antioxidant responses in various algal species under heavy metal stress. This broader context will highlight the unique aspects of Chlamydomonas reinhardtiiin coping with cobalt-induced stress.

The introduction section has been extended accordingly.

2. How you selected CoCl₂ concentrations (up to 125 µM). Were these concentrations chosen based on preliminary toxicity assays, and do they reflect natural exposure scenarios?

The detailed explanation was added to the materials and methods section, together with the supplementary file with the results of preliminary experiments.

Considering natural exposure, cobalt is usually rare in freshwater environments, except for the situations where mining, manufacturing, or other activities may produce locally elevated concentrations. Judging on the data found in the literature, usually the Co concentration in water does not exceed 1 mM. In groundwaters from Zahedan Co concentration was about 2,9-3,5 mM. In mining-influenced streams in Idaho it reached up to 15 mM. Most highly contaminated sites in Canada reported Co concentrations were up to 35 mM.

Atashi, H., Sahebi-Shahemabadi, M., Mansoorkiai, R., & Spaili, F. A. (2009). Cobalt in Zahedan drinking water. J. Appl. Sci. Res5(12), 2203-2207.

Mebane, C. A., Eakins, R. J., Fraser, B. G., & Adams, W. J. (2015). Recovery of a mining-damaged stream ecosystem. Elementa, 3, 000042.

Stubblefield, W. A., Van Genderen, E., Cardwell, A. S., Heijerick, D. G., Janssen, C. R., & De Schamphelaere, K. A. (2020). Acute and chronic toxicity of cobalt to freshwater organisms: using a species sensitivity distribution approach to establish international water quality standards. Environmental Toxicology and Chemistry, 39(4), 799-811.

However, we wanted to test a Co concentration range up to concentrations causing severe toxicity. It may be of use, as algae are thought as candidates for wastewater treatment, therefore the studies concerning HM toxicity and tolerance in these organisms have a potential of application.

3. The decrease in NPQ efficiency is an interesting finding. However, the interpretation that cobalt may inhibit chlororespiration activity is speculative. Could the authors provide stronger evidence, or otherwise phrase this more carefully as a hypothesis requiring future testing?

This part of discussion was rephrased.

4. The differential effects on SOD (decrease) vs CAT/APX (increase at certain concentrations) are interesting. The discussion should explain on possible mechanistic reasons why SOD is specifically inhibited by Co²⁺ in this model.

The discussion section has been extended to provide plausible explanation.

5. For proline, the lack of accumulation is notable, given its common role in stress tolerance. The authors could expand on why C. reinhardtii may not use proline under cobalt stress.

The discussion concerning proline has been extended.

Minor Comments

6. Authors thoroughly revise the manuscript to avoid grammatical and typo errors

The manuscript has been revised and corrected.

7. Lines 31-33:  “the other do not” grammatical error; should be “others do not.”

This part of the introduction was changed according to the suggestion given by another reviewer, therefore this part of the sentence no longer exists there.

8. Lines 37-41: (…)  Sentence is too long and difficult to follow. Spilt it into simple sentence. (…) Add relevant reference here

The sentence was split and the suggested references were added.

9. Line 43: (…) Check the spelling “thioyl” or “thiol” is more correct in biochemical context.

Spelling was corrected.

Lines 61-62: (…)  “disturb” is not correct, use “disrupt.”

The word was changed as suggested.

10. Lines 88- 95 (Aim of study) (…) Very long sentence, difficult to explain. Revise it

This section was rewritten.

11. Lines 106- 107:...but the effect did not turn out to be statistically significant.  “did not turn out to be.” is not clear. Improve it as: Example: ...but this difference was not statistically significant.

The text was changed accordingly.

12. Lines 389–390:  For the sake of the experiments, algae were inoculated to provide an initial Chl a + b concentration of 0.5 µg/ml... “For the sake of” is not clear. Example: For all experiments, cultures were inoculated at an initial chlorophyll a + b concentration of 0.5 µg/ml.

The text was changed accordingly.

13. Line 429-31 (…) Grammar: “has cell wall” should be “has a cell wall.”

The lacking article was added.

14. Lines 445–446; As it was expected, excessive Co caused inhibition of C. reinhardtii growth and decrease in total Chl content compared to the control. “As it was expected” unnecessary; conclusion should be concise

The conclusion section has been rewritten, according to the suggestion of another reviewer, therefore the text is now different.